# Estimation of Greenhouse Gas Emissions of Petrol, Biodiesel and Battery Electric Vehicles in Malaysia Based on Life Cycle Approach

**Shoki Kosai** [1] ![ID], **Sazalina Zakaria** [2,*], **Hang Seng Che** [3] ![ID], **Md Hasanuzzaman** [3] ![ID], **Nasrudin Abd Rahim** [3] ![ID], **Chiakwang Tan** [3], **Radin Diana R. Ahmad** [2], **Ahmad Rosly Abbas** [2], **Katsuyuki Nakano** [4] ![ID], **Eiji Yamasue** [5] ![ID], **Wei Kian Woon** [6] and **Ammar Harith Ahmad Amer** [2]

1   Global Innovation Research Organization, Ritsumeikan University, Kusatsu 525-8577, Shiga, Japan; kosai@fc.ritsumei.ac.jp
2   Built Environment and Climate Change Unit, Generation & Environment Department, TNB Research Sdn. Bhd., Kajang 43000, Malaysia; diana.ahmad@tnb.com.my (R.D.R.A.); rosly.abbas@tnb.com.my (A.R.A.); ammar.amer@tnb.com.my (A.H.A.A.)
3   Higher Institution Centre of Excellence (HICoE), UM Power Energy Dedicated Advanced Centre (UMPEDAC), Level 4, Wisma R&D, University of Malaya, Jalan Pantai Baharu, Kuala Lumpur 59990, Malaysia; hsche@um.edu.my (H.S.C.); hasan@um.edu.my (M.H.); nasrudin@um.edu.my (N.A.R.); cktan@um.edy.my (C.T.)
4   College of Policy Science, Ritsumeikan University, Ibaraki 567-8570, Osaka, Japan; nakanok@fc.ritsumei.ac.jp
5   Department of Mechanical Engineering, College of Science and Engineering, Ritsumeikan University, Kusatsu 525-8577, Shiga, Japan; yamasue@fc.ritsumei.ac.jp
6   Corporate Strategy & Sustainability, Strategy and Ventures Division, Tenaga Nasional Berhad, Kuala Lumpur 50470, Malaysia; wkw@tnb.com.my
*   Correspondence: sazalina.zakaria@tnb.com.my

**Abstract:** A steady rise in the ownership of vehicles in Malaysia has drawn attention to the need for more effective strategies to reduce the emissions of the road transport sector. Although the electrification of vehicles and replacing petrol with biofuel are the strategies being considered in Malaysia, these strategies have yet to be fully evaluated from an environmental perspective. In this study, a life cycle assessment was conducted to compare the greenhouse gas emissions of different types of transportation means (passenger cars, two-wheelers (motorbikes), and buses) with several types of powertrains (petrol, biodiesel, electricity) based on multiple lifecycle stages in Malaysia. The impact of considering land use change for the biodiesel production in the LCA was also considered in this study. It was found that the transition from internal combustion engine vehicles fueled by petrol to electric vehicles would reduce the greenhouse gas emission for passenger cars, two-wheelers, and buses. However, because the greenhouse gas emissions of biodiesel-fueled vehicles are higher than those of petrol-fueled vehicles, even without considering land use change, the results indicate that the transition from a 10% to 20% biofuel blend, which is a current strategy in Malaysia, will not result in a reduction in greenhouse gas emissions for the transport sector in Malaysia.

**Keywords:** life cycle assessment; biodiesel; land-use change; palm oil; electric vehicle; ASEAN

## 1. Introduction

In Malaysia, the energy sector is projected to continue to be the largest contributor to greenhouse gas (GHG) emissions in 2030. According to the Third National Communication (NC3) of Malaysia [1], the largest contributor to increasing GHG emissions in the energy sector is the transportation industry, with emissions expected to increase from 64,387 Gg $CO_2$eq in 2014 to 114,840–116,700 $CO_2$eq in 2030. Since most of the energy use in the transport industry is from road transportation, an effective approach to reducing the emissions of road transport is required.

As of 2018, the highest rate of motorisation among ASEAN countries has been in Malaysia at 4 times the continental average, or 439 vehicles per 1000 inhabitants. The first and most pressing challenge in the effort to reduce emissions is to address the transport energy use.

The electrification of vehicles, including automobiles, two-wheelers and buses, is one of the main strategies to mitigate GHG emissions in the road transport sector. As a result of this global trend away from electrification, it is expected that battery electric vehicles (BEVs) will account for approximately 50% of the global automobile stocks in 2050 [2]. The BEV industry has yet to gain much traction in Malaysia and is unlikely to do so until further technological gains are realised and the prices of BEVs fall. Because Malaysia is a net oil exporter with relatively low fuel prices, it is expected to take longer to reach parity.

In addition to the electrification of the road transport sector of Malaysia, the use of biofuels is another strategy intended to mitigate GHG emissions [3]. Biodiesel is considered a sustainable and viable alternative source to petrol in Malaysia, one of the largest producers of palm oil [4]. Based on National Biofuel Policy in 2006 [5], a blend of 90% petroleum diesel 10% of biofuels, known as B10 biodiesel, has been used in the transportation sector since 2019. The Malaysian government is said to be considering a new biodiesel strategy: the use of B20, which includes 20% bioenergy produced from palm oil.

In this study, a life cycle assessment (LCA) was completed to evaluate the action for mitigating GHG emissions in the road transport sector of Malaysia. A comparative environmental life cycle assessment has been applied to evaluate the differences in GHG emissions between Internal combustion engine vehicles (ICEVs) and BEVs in Poland [6], the Czech Republic [7], Spain [8], Lithuania [9], Norway [10], Canada [11], Sweden [12], China [13], Hong Kong [14], the United States [15], Portugal [16], and Japan [17] and between petrol-powered vehicles and biodiesel-powered vehicles in Brazil [18] and France [19]. While Malaysia [20] and Indonesia [21] have been considered using a similar LCA, the number of the relevant studies in the Association of South-East Asian Nations (ASEAN) are limited.

A common trend in many of the studies is that the GHG emissions of ICEVs are larger than those of BEVs in all of the countries assessed. Since BEVs are likely to be introduced in Malaysia in the near future, it is necessary to confirm whether the GHG emissions in Malaysia is similar to that in other countries.

Unlike the relationship of GHG emissions between BEVs and ICEVs, that between biodiesel-powered vehicles and petrol-powered vehicles varies. For instance, Messagie et al. [22] reported that GHG emissions of biodiesel-powered vehicles using B100 are larger than those of petrol-powered vehicles, whereas de Souza [18] reported the opposite trend. Similarly, different LCA trends of GHG emissions on biodiesel production have been also seen (e.g., [23,24]). Puricelli et al. pointed out in a recent review article the main problem with biofuel LCAs is the inconsistency due to the inclusion or exclusion of land use change in the system boundary: this results in a significant difference in the findings for GHG emissions [25]. From this perspective, it is important to evaluate the GHG emissions of biodiesel-powered vehicles in Malaysia and also analyse the impact of considering land use change on the findings.

Thus, the objective of this study is to estimate the GHG emissions of petrol, biodiesel and battery electric vehicles, including passenger cars, two-wheelers and buses, in Malaysia based on the life cycle approach, both with and without the consideration of land use change. We expect that the results of this study will serve as indicators of promising strategies to mitigate GHG emissions in the road transport sector.

## 2. Materials and Methods

In this study, our aim is to compare the GHG emissions of different types of transportation means in Malaysia. A comparative LCA on an attributional and process basis was conducted to identify the environmental impacts of vehicle and fuel cycles. The sub-

jects of LCA are differentiated into the following types of transportation means and types of powertrains:

- Internal combustion engine vehicle fuelled by petrol ($ICEV_{petrol}$);
- Internal combustion engine vehicle fuelled by biodiesel ($ICEV_{biodiesel}$);
- Battery electric vehicle (BEV);
- Plug-in hybrid vehicle (PHEV);
- Internal combustion engine motorcycle fuelled by petrol (ICE2W);
- Battery electric motorcycle(E2W);
- Internal combustion engine bus fuelled by diesel ($ICE-Bus_{diesel}$);
- Internal combustion engine bus fuelled by biodiesel ($ICE-Bus_{biodiesel}$);
- Battery electric bus (E-bus).

Considering the cradle-to-grave system for each material and type of fuel, a life cycle inventory (LCI) analysis, including foreground and background data, was carried out under the International Organization for Standardization (ISO) 14040 standard [26]. The life cycle inventory database known as IDEA 2.2 [27] was used in this study to calculate GHG emissions, as shown in Supplementary Materials.

The following life cycle phases were considered in this study for each case: the production phase, operational phase, maintenance phase and disposal phase. Neither the transport infrastructure (e.g., construction of road, fuel station) nor the recovery of vehicle parts through recycling were considered in this study. The recycling of the lithium-ion batteries has yet to be fully commercialized in the recycling system of Malaysia. An overview of the system boundary is presented in Figure 1.

**System boundary**

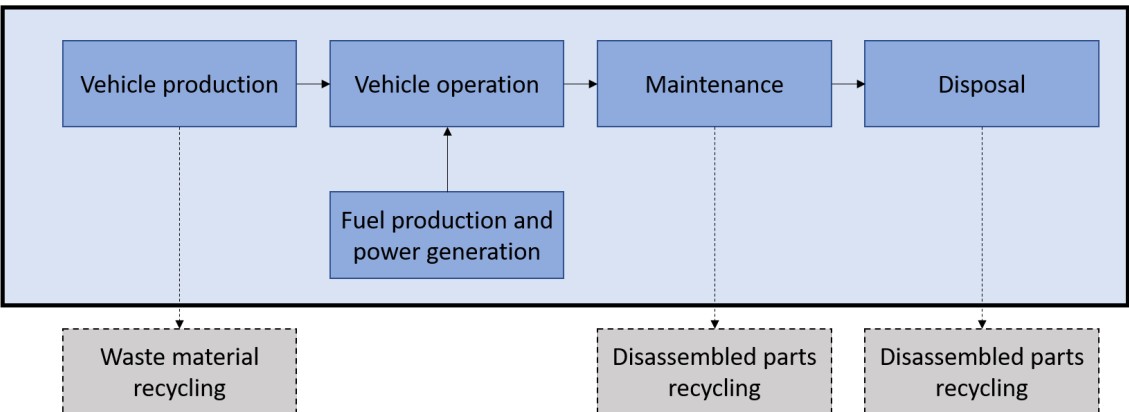

**Figure 1.** System boundary of this study.

The functional unit is greenhouse gas emissions per 1 kilometre (km) driven under average Malaysian conditions based on the Low Carbon Mobility Blueprint (LCMB) of Malaysia. The vehicle size for this study is an ordinary passenger automobile with a weight of 1500 kg with 2000 cc. The study is based on the motorcycle sizes on 125 cc of ICE2W, which has the highest market share in Southeast Asia countries [28]. The bus size considered in this study is an ordinary commercial bus with a weight of 8000 kg [29]. In this study, the chassis designed for the PHEV and BEV, E2W and E-bus is the same as that developed for the ICEV, ICE2W and ICE-Bus, respectively. The only difference is in the powertrain and the fuel supply.

### 2.1. Production Phase

The emissions produced for the production phase of passenger car, motorcycle and bus were calculated for the following: (1) the chassis, (2) the engine and transmission for ICE vehicles and (3) the inverter, motor and a lithium-ion battery for electric vehicles. It was assumed that the chassis parts (the body, tires, interiors, etc.) are identical for the ICEV,

BEV and PHEV in passenger cars for the ICE2W and E2W motorcycles and ICE-bus and E-bus in buses.

A set of inventory data for each vehicle was established using the LCA software "MiLCA" [30], which is a complete inventory data of ICE vehicle production. These inventory data were directly employed to consider the production of the ICEV, ICE2W and ICE-bus. In the case of electric vehicles (BEV, E2W, E-bus), it was necessary to exclude the parts of engine and transmission from this inventory data for ICE vehicles and then add the inverter, motor and lithium-ion battery. Notably, the same engine and transmission used in the ICEV were assumed to be installed in the PHEV.

The inventory data of the lithium-ion battery (characterized by a composite cathode material of lithium manganese oxide ($LiMn_2O_4$) and lithium nickel manganese cobalt oxide $Li(Ni_xCo_yMn_{1-x-y})O_2$), inverter and the motor were taken from the study by [31–33], respectively. The specifications of these representative BEVs are provided in Table 1.

**Table 1.** Specifications of the lithium-ion battery, inverter and motor for the BEV.

| Parts | Weight (kg) | Capacity | Source |
|---|---|---|---|
| Lithium-ion battery | 253 | 26.6 kWh | [31] |
| Inverter | 15.1 | 135 kW | [32] |
| Motor | 58.3 | 135kW | [33] |

Based on the development of inventory data for BEV production, the input data for the lithium-ion battery, inverter and motor were rescaled so that it was applicable for the PHEV, E2W and E-bus. The rescaled values are listed in Table 2. The size of the battery of the E-bus was based on the BYD K9 bus.

**Table 2.** Size of the lithium-ion battery, inverter and motor for PHEV, E2W and E-bus.

| Vehicle Type | Rescaled Rate | Source |
|---|---|---|
| PHEV | Lithium-ion battery: 8.8 kWh/26.6 kWh<br>Inverter: 15.8 kW/135 kW<br>Motor: 15.8 kW/135 kW | [30] |
| E2W | Lithium-ion battery: 0.9 kWh/26.6 kWh<br>Inverter: 6 kW/135 kW<br>Motor: 1 kW/135 kW | [31,32] |
| E-bus | Lithium-ion battery: 324 kWh/26.6 kWh<br>Inverter: 200 kW/135 kW<br>Motor: 200 kW/135 kW | [29] |

### 2.2. Operational Phase

The first step was to determine the tank-to-wheel and battery-to-wheel fuel economy for each vehicle. The fuel of the vehicles differed: ICEVs consume petrol and biodiesel (B10), ICE2Ws consume petrol and ICE-buses consume biodiesel (B10). PHEVs have both a petrol and an electric mode. The fuel economy of the ICEV, the BEV, the ICE2W and the ICE bus was taken from the report by LCMB. The fuel economy for the E2W is the mean value of the three representative E2Ws reported in [34,35]. The E-bus fuel economy is the mean value of two representative E-buses reported in [36,37]. The summary of the tank-to-wheel and battery-to-wheel fuel economy applied in this study is presented in Table 3.

**Table 3.** Fuel economy for each vehicle.

| Vehicle | | Fuel Economy | | |
|---|---|---|---|---|
| | | Petrol (km/L) | Biodiesel (km/L) | Electricity (km/kWh) |
| Passenger car | ICEV (petrol) | 11.78 | - | - |
| | ICEV (diesel) | - | 8.71 | - |
| | BEV | - | - | 5.67 |
| | PHEV | 11.78 | - | 5.67 |
| Motorcycles | ICE2W | 46.23 | - | - |
| | E2W | - | - | 25 |
| Bus | ICE-bus | - | 3.41 | - |
| | E-bus | - | - | 1.5 |

In this study, the GHG emissions through the production and combustion of fuels and electric power generation were calculated as follows:

(1) The system boundary for petrol and diesel production is from resource extraction to combustion via service stations. The emission factors of these fuels in MiLCA are specified with the amount of emissions from 1 MJ of fuel combustion (e.g., petrol: 0.081 kg-$CO_2$/MJ, diesel: 0.076 kg-$CO_2$/MJ). Therefore, the heating value of fuels (petrol: 0.03 L/MJ, diesel: 0.026 L/MJ) [38] was used to convert those factors into the amount of emissions from 1L of fuel combustion (e.g., kg-$CO_2$/L).

(2) Biodiesel production is mainly based on palm oil in Malaysia. The inventory data of biodiesel production in Malaysia was developed using the data from existing studies [39–42]. This study considered land-use change during the process. Inventory data were taken from an earlier study [43]. Traction biodiesel (B10) in Malaysia contains 10% of bioenergy produced using palm oil.

(3) Peninsular Malaysia electricity generation mix data in 2018 were used (coal: 53%, natural gas: 42%, hydro: 4%, RE: 1.0%, and diesel: 0.1% [44]). The system boundary of electric power generation was from energy resource extraction to end-user use via conversion from fuel to electricity and transmission/distribution.

The total mileage under the operational phase depends on the lifespan of each vehicle. It is assumed that the PHEV runs under the petrol mode at half mileage and the electric mode at rest [45]. In this study, the interval for lithium-ion battery replacement is selected as the functional mileage, as described in the following section.

*2.3. Maintenance Phase*

For vehicle maintenance, it is necessary to replace some parts. It is widely accepted that lithium-ion battery replacement results in a significant amount of GHG emissions (see [46,47]). In this study, the lithium-ion battery replacement mileage depends on the vehicle type. That of the BEV was cited from the warranty distances announced by several automobile companies [46,48]. That of the E2W follows the India case [49]. The battery replacement of the E-bus was estimated at 620,000 km, based on the lifetime of the battery [50] and average annual mileage of the bus in Malaysia. The replacement of tires and the lead-acid battery was also considered, referring to an earlier report [46]. The summary of maintenance for each vehicle is presented in Table 4. The interval for lithium-ion battery replacement was used as functional mileage.

**Table 4.** Maintenance stage for each vehicle.

| Parts | Replacement Interval (km) | Applied Vehicle |
|---|---|---|
| Tire | 40,000 | ICEV, BEV, PHEV, ICE2W, E2W, ICE-bus, E-bus |
| Lead-acid battery | 50,000 | ICEV, BEV, PHEV, ICE2W, E2W, ICE-bus, E-bus |
| Lithium-ion battery | 160,000<br>80,000<br>80,000<br>620,000 | BEV<br>PHEV under the electric mode<br>E2W<br>E-bus |

*2.4. Disposal Phase*

The disposal phase included disassembly, shredding, sorting and landfilling in this study. It is represented as the amount of emissions per 1 kg of vehicle parts. The body, interior, exterior parts were considered for ICEV in the MiLCA and this emission factor was applied to the other assessed vehicles.

**3. Results**

The GHG emissions of the passenger cars, two-wheelers, and buses in Malaysia are presented in Figure 2.

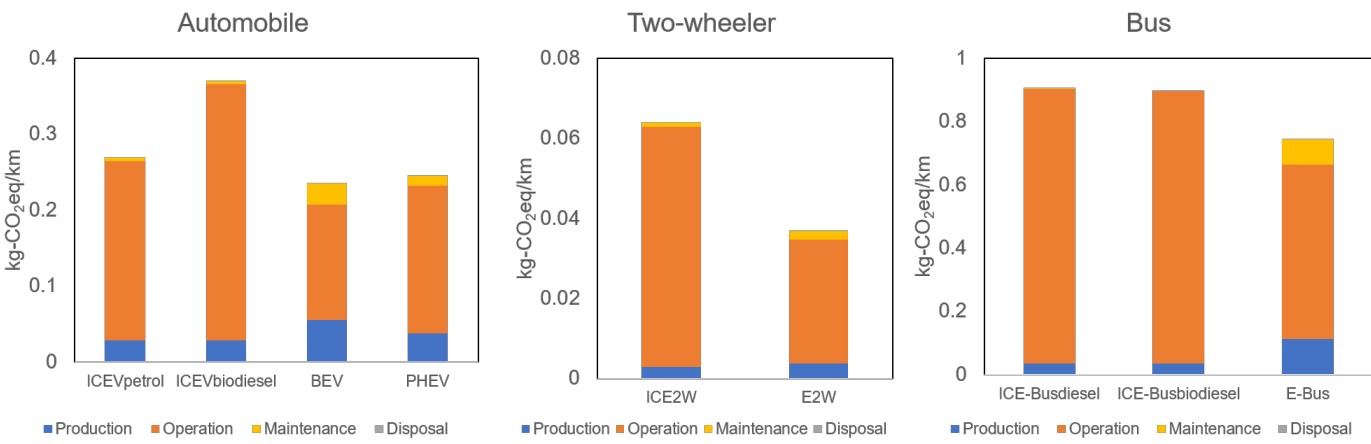

**Figure 2.** GHG emissions of passenger cars, two-wheelers and buses in Malaysia.

The calculated GHG emissions of passenger cars were in the increasing order of BEV (0.236 kg-$CO_2$eq/km), PHEV (0.245 kg-$CO_2$eq/km), ICEV$_{petrol}$ (0.269 kg-$CO_2$eq/km) and ICEV$_{biodiesel}$ (0.370 kg-$CO_2$eq/km).

Under the production phase, the GHG emissions of the battery electric vehicle (BEV) were the highest among passenger cars. This is attributed to the high material consumption of the lithium-ion battery. The GHG emissions associated with BEV production are about 1.9 times higher than those associated with ICEV production. The same trend has been reported in the existing literature, with the $CO_2$ emissions of ICEV and production almost double that of BEV production (see 5600 kg-$CO_2$ for ICEV and 12,000 kg-$CO_2$ for BEV in [46]). The lithium-ion battery contains many materials and metals which are not used in the production of ICEVs. Cobalt production, lithium, copper and aluminium contribute to the increased GHG emissions of the lithium-ion battery.

The operational phase accounts for the most significant share of GHG emissions regardless of vehicle type among the passenger cars (91% of ICEV$_{biodiesel}$, 88% of ICEV$_{petrol}$, 64% of BEV and 79% of PHEV). This study used the battery replacement interval as the functional mileage, but this share is greater under lifetime mileage. The ICEV had the

highest GHG emissions under the operational phase among the passenger cars. The main contributor is tailpipe emissions from petrol combustion rather than petrol production.

Under the maintenance phase, the BEV and PHEV had higher GHG emissions than the ICEV. The replacement of the lithium-ion battery is the major contributor. This trend was supported by the findings of an earlier study [51]. The impact of the replacement of tires and the lead-acid battery is negligibly small under the maintenance phase.

The GHG emissions trend of two-wheelers and buses is quite similar. The GHG emissions of the ICE2W ($0.064$ kg-$CO_2$eq/km) are approximately 1.8 times greater than those of the E2W ($0.035$ kg-$CO_2$eq), while the GHG emissions of the ICE-bus ($0.91$ kg-$CO_2$eq/km) are approximately 1.2 times greater than the E-bus ($0.74$ kg-$CO_2$eq/km).

The GHG emissions of lithium-ion battery production used for the E2W is approximately 135 kg-$CO_2$eq, which is equivalent to almost 45% of the emissions for the production of the E2W, while the GHG emissions for the lithium-ion battery in the E-bus is approximately 50,000 kg-$CO_2$eq, which is equivalent to almost 70% of emissions for the production of the E-Bus. Among the two-wheelers, the operational phase accounts for the most significant share of GHGs regardless of vehicle type (94% of the ICE2W emissions and 84% of the E2W emissions). This was also found for the buses (91% of the ICE-bus emissions and 74% of the E-bus emissions).

Among the three types of transportation, the GWP of the ICE-bus is 3.4 times greater than the ICEV$_{petrol}$ and 14 times greater than the ICE2W. In the case of the E-bus, the GWP is 3.2 times greater than that of the BEV, and 20 times greater than the E2W. Weight and fuel economy have a large effect on the GHG emissions. The battery replacement interval was considered the functional mileage in this study. Each transportation means has a different battery replacement interval (BEV: 160,000 km, E2W: 80,000 km, and E-bus: 620,000 km). Therefore, the functional unit of production, maintenance and disposal stages is different for the passenger car, motorcycle and bus.

## 4. Discussion and Conclusions

The focus of this study is on electrification in the road transport sector and the use of biofuels and their potential for reducing the GHG emissions of the transport sector in Malaysia. A life cycle approach was used to evaluate the GHG emissions of petrol, biodiesel and battery electric vehicles, including passenger cars, two-wheelers and buses, in Malaysia. According to our results, the GHG emissions of BEVs are smaller than those of ICEVs fuelled by petrol, while those of ICEVs fuelled by biodiesel are larger. Based on the results, electrification and the use of biofuels in the road transport of Malaysia are further discussed from the perspective of the type of biofuel and the land use change associated with biodiesel production and the electricity mix in Malaysia.

The type of biodiesel and land use change need to be carefully considered in the LCA of ICEVs fuelled by biodiesel. The B10 applied in this study contains 10% bioenergy produced from palm oil and 90% diesel and is currently in use in Malaysia. The Malaysian government has a new biodiesel strategy using B20, with 20% bioenergy produced from palm oil. Ultimately, it is presumed that B100 will also be considered. With this in mind, and in keeping with the global trend, the impact of the land use change in the process of palm oil production was considered in the LCA. It should be noted that most of the existing LCA studies of palm oil production in Malaysia exclude the impact of land use change (e.g., [42]), and there is some controversy about the inclusion of land use change. For this reason, the impact of the difference in the type of biodiesel both considering and not considering land use change on the GHG emissions of ICEVs was analysed in this study, as presented in Figure 3.

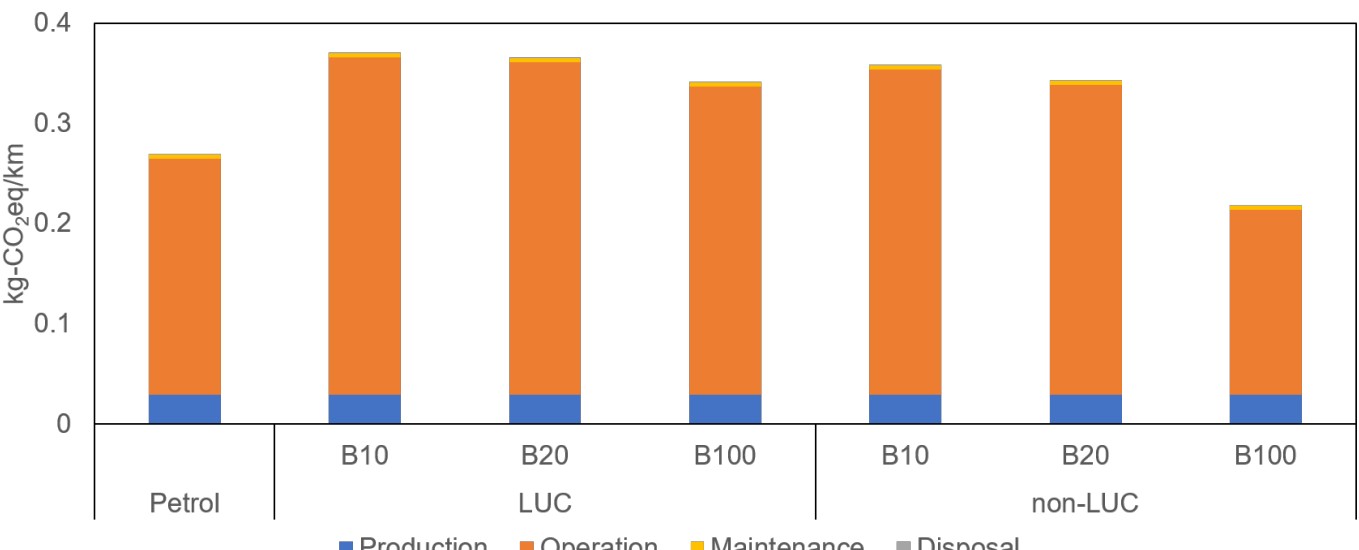

**Figure 3.** GHG emissions of ICEV considering different types of biodiesels and land use changes (LUC).

In the case where land use change is considered, negligible difference was found for the GHG emissions during the operational stage between B10 and B20. In terms of the transition from B10 to B20, the results indicate that the Malaysian strategy to transition to B20 is not likely to be effective. This is because the GHG emissions per litre of biodiesel production based on Malaysian palm oil and combustion (2.68 kg-$CO_2$eq/L) are similar to those for diesel production and combustion (2.98 kg-$CO_2$eq/L). Thus, the GHG emissions in the ultimate case of B100 are even larger than those of ICEV$_{petrol}$.

In the case where land use change is not considered, the GHG emissions of ICEVs fuelled by B10 and B20 are still larger than ICEVs fuelled by petrol. This is because the fuel economy of petrol-fuelled ICEVs is higher than that of diesel-fuelled ICEVs in Malaysia. Thus, as was also found when considering land use change, the findings of this study indicate that the Malaysian strategy of a transition from B10 to B20 will not achieve the desired goal of reducing the GHGs of the transport sector. It should be noted, however, that the GHG emissions of ICEVs fuelled by B100 are lower than those of ICEVs fuelled by petrol. Our findings indicate that at least 68% of the bioenergy produced by palm oil needs to be mixed with diesel even when land use change is not considered in the analysis for ICEVs fuelled by biodiesel to be favourable.

Land use change contributes to 40% of the total GHG emissions associated with biodiesel production. Whether this component is included or excluded in estimating GHGs is a serious matter when considering the use of biodiesel as a strategy since a high ratio of bioenergy is produced by palm oil. In some articles on ASEAN countries, it has been argued that the continued land use practice in the palm oil industry serves to minimize land use changes and biodiversity losses in Malaysia [52], Thailand [53]. That is, the land use change associated with the industry was not considered. However, in a recent study on the land use change and biodiversity loss caused by palm oil production in the ASEAN region [54,55], the findings strongly support the argument for considering the land use change, as is the global standard. In some studies focusing on Indonesia, one of the ASEAN countries, land use change in the LCA of palm oil is included, following this global notion (e.g., [56]).

It must be noted that the palm oil industry is vital to the economy of Malaysia. However, negative perceptions which stem from the environmental burdens and labour issues associated with palm oil production have led to anti-palm oil campaigns and trade regulations in developed countries, particularly in the European Union (EU) [57]. Although this topic is politically sensitive due to the historic background of the palm oil industry [58], policymakers in key stakeholder agencies in Malaysia are now required to standardize

the framework of palm oil production for environmental assessment based on sustainable plantation practices founded in scientific knowledge.

This study was based on the electricity mix of Peninsular Malaysia in 2018, and the share of fossil fuels in 2018 in Peninsular Malaysia was found to be remarkably high (95%). It was found that increasing the share of renewable energy in the electricity mix would reduce the GHG emissions under the operational phase for the BEV, E2W and E-bus. As such, the electricity mix in 2040 in Peninsular Malaysia projected by the Malaysian electricity procurement services provider, Single Buyer was used to evaluate the potential decrease in the GHG emissions in the case that electric vehicles are adopted.

The amount of fuel used for each power source needs to be calculated from electricity generation mix data. The conversion process from the share of electricity generation mix to fuel input by power source was achieved by considering the electricity loss through the transmission and distribution, electricity used by plants and the conversion efficiency. It is assumed that the conversion process in 2040 is the same as that in 2018. The data for electricity consumption in 2018 were taken from the National Energy Balance 2018 [59]. The data of transmission and distribution loss in 2018 were taken from the Annual report for 2018 of TNB [60]. The data of electricity used by the plant and fuel conversion efficiency by a power source in 2018 were taken from Performance and Statistical Information on the Malaysian Electricity Supply Industry 2018 of Suruhanjaya Tenaga [44].

The GHG emissions of electric vehicles in 2018 and in 2040 are presented in Figure 4. The change in the electricity generation mix from 2018 to 2040 in Peninsular Malaysia contributes to a decrease in GHG emissions by 19.8% in the case of the BEV, 25.9% in the case of the E2W and 22.7% in the case of the E-Bus. The decreasing share of coal power is the main contributor to the decrease in GHG emissions from 2018 (53%) to 2040 (26%). Thus, at least under the operational phase, the transition from the ICEV to the EV is highly recommended in Malaysia from the perspective of GHG emissions because of the increase in the share of renewable energy.

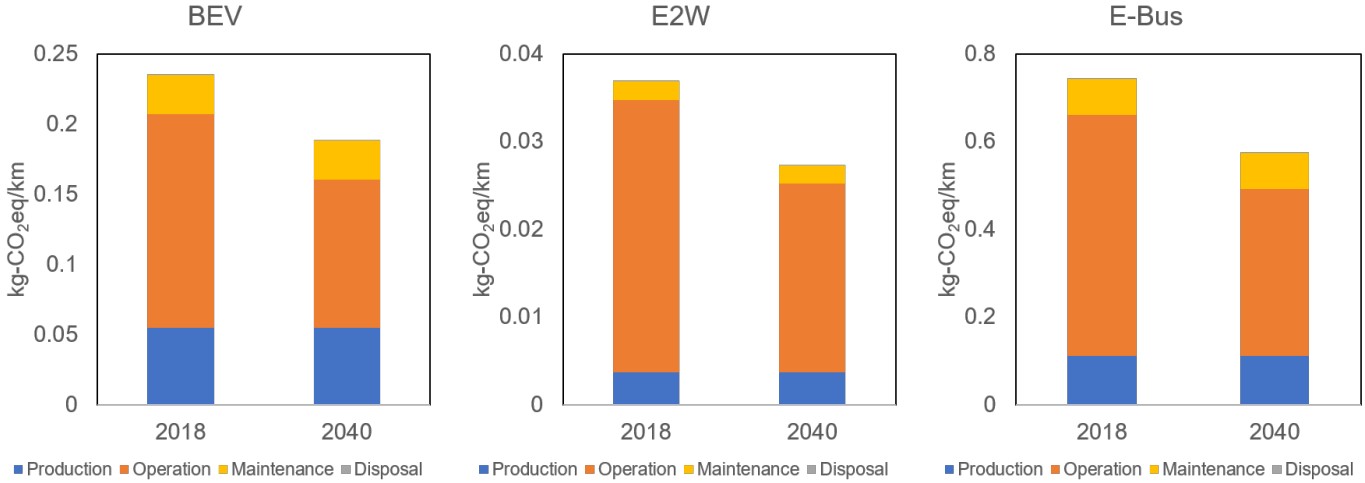

**Figure 4.** GHG emissions of electric vehicles in 2018 and in 2040 considering the change in electricity mix in Malaysia.

It should be noted that the transition of the material components from 2018 to 2040 has not been considered in this study. In this analysis, the same type and amount of material for lithium-ion battery production in 2018 is applied to the case in 2040. The projection and development of lifecycle inventory data for lithium-ion battery production in 2040 contain various uncertainties at the current stage. For instance, it is projected that the energy density of lithium-ion batteries will increase in the future, which will result in lighter batteries. It is also possible that materials with a greater or lesser impact on the environment than the current materials will be used. Since so much depends on the type of technological

innovations to come, it is difficult to identify which material used for the lithium-ion battery will be affected and how much the weight of the corresponding material will change. At least, the results of LCA presented in this report can be considered as business-as-usual.

The end-of-life treatment of EV batteries is also not considered in this analysis. The technology for recycling the lithium-ion battery is still at the research stage, and it has yet to be fully commercialized across the globe. In addition, although many suppliers deal with the parts remanufactured from end-of-life vehicles, these are not properly managed or recycled in Malaysia because of the inadequate regulation system [61]. As the technology for second-life batteries and the recycling of EV batteries develops, the life cycle emissions of EV batteries are likely to be reduced. From this perspective, we can say that the results of the LCA presented in this study can be considered a worst-case scenario. The integration of the recycling stage in the current system boundary should be considered in future studies to re-examine the impact of EV adoption.

**Supplementary Materials:** The following supporting information can be downloaded at: https://www.mdpi.com/article/10.3390/su14105783/s1, Table S1: Inventory data.

**Author Contributions:** Conceptualization, S.K., H.S.C. and M.H.; methodology, S.K., K.N. and H.S.C.; software, S.K., K.N. and E.Y.; validation, C.T. and H.S.C.; formal analysis, S.K., K.N., E.Y. and C.T.; investigation, K.N. and E.Y.; resources, C.T. and M.H.; data curation, S.K., M.H. and S.Z.; writing—original draft preparation, S.K. and S.Z.; writing—review and editing, H.S.C., R.D.R.A. and A.H.A.A.; visualization, S.K.; supervision, N.A.R., A.R.A. and W.K.W.; project administration, N.A.R. and A.R.A.; funding acquisition, S.K. and W.K.W. All authors have read and agreed to the published version of the manuscript.

**Funding:** This paper was partly supported by research fund from KAKENHI Grants (20K20013); by the Ritsumeikan Global Innovation Research Organization(R-GIRO), Ritsumeikan University; and a research project named "Environmental and Health Impacts of Growth in the Electric Vehicle (EV) Industry to the Country and the Electricity Supply Industry".

**Institutional Review Board Statement:** Not applicable.

**Informed Consent Statement:** Not applicable.

**Data Availability Statement:** The data presented in this study are available in the Supplementary Materials.

**Conflicts of Interest:** The authors declare no conflict of interest. The funders had no role in the design of the study; in the collection, analyses, or interpretation of data; in the writing of the manuscript; or in the decision to publish the results.

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
