# Peer review of "Estimation of Greenhouse Gas Emissions of Petrol, Biodiesel and Battery Electric Vehicles in Malaysia Based on Life Cycle Approach"

_sustainability, doi:10.3390/su14105783_

Round 1
Reviewer 1 Report
The authors provided a typescript on the topic Estimation of greenhouse gas emissions of petrol, biodiesel and battery electric vehicles in Malaysia based on life cycle approach.
The research method should be completed in the summary.
When entering numerical values, the comma should be replaced with a dot. In the introduction, the authors provide some interesting facts about the amount of fuel consumption and engine use in European countries - however, these data are not supported by statistical data and no references. The introduction needs to be thoroughly revised. I suggest that a table be prepared and the results of other authors' research together with references should be cited in one place. See lines: 63-91.
The authors write that they have conducted the LCIA for 10 different vehicles covering the phases: the production phase, operational phase, maintenance phase and disposal phase.
Neither Table 1 nor Table 2 is an inventory table. At the production stage of, for example, an engine, we need information on the amount of materials, raw materials, media, etc. used. In its current form, we cannot speak of an LCA analysis carried out at the production stage. I would also like to add that the data is not complete - there is no source. Similarly, each subsequent phase of the life cycle requires correction.
For the last phase, it is very questionable that "The amount of GHG emissions under the disposal phase for ICEVs was cited from the LCA software MiLCA, which include disassembly, shredding, sorting, and landfilling." Does the program determine? It calculates the program? Information that is stylistically and scientifically misrepresented.
The results do not include analysis. I am quoting: “The GHG emissions of the passenger cars, two-wheelers, and buses in Malaysia are presented in Figure 2. The calculated GHG emissions of passenger cars was in the increasing order of BEV (0.236 kg-CO2eq/km), PHEV (0.245 kg-CO2eq/km), ICEVpetrol (0.269 kg-CO2eq/km), and ICEVbiodiesel (0.370 kg-CO2eq/km).”
I would also like to add that the figures in the figures are not legible. Bar charts do not represent reliable and accurate data.
I recommend the authors to read the sample publications:
- https://doi.org/10.3390/ma14164556
- https://doi.org/10.1016/j.jclepro.2018.12.058
- https://doi.org/10.1016/j.jclepro.2018.08.145
- https://doi.org/10.1016/j.trd.2018.03.005
- https://doi.org/10.1016/j.apenergy.2018.09.139
The manuscript in its present form requires thorough improvement.
Author Response
We wish to express our appreciation to the Reviewers for his or her insightful comments, which have helped us significantly improve the paper.
Please see the attached file.

Reviewer 2 Report
Proposed paper targets two important topics: LCA and its relationship with the transportation sector in Malaysia. It provides some estimates about the GHG impacts including biofuels and, in particular, those obtained from palm oil, including Land use Change considerations. Interesting the comparison performed with electric transportation, and the inclusion of two wheels and collective vehicles.
I consider the paper worth publishing because it highlights how the expected GHG in the transportation sector in Malaysia is likely to get worse if current policies remain in place.
Prior to publication I recommend some changes:
- References indications are missing in the text, they must be indicated.
- L 143. There is a gap since the reference is not indicated here.
- Tab 1 and 2. Source is missing.
Check for typos in the text
Author Response

(The authors gave the same response as above.)

Reviewer 3 Report
Author did commandable work on estimation of Greenhouse Gas emission from various type of transportation systems. The novelty of the work clearly explained. The structure of the manuscript and discussion also good. I recommend for the publication.
Author Response
Thank you very much for reviewing our paper and giving us "accept"!
Round 2
Reviewer 1 Report
The authors have thoroughly revised their typescript. Nevertheless, there are still some things that need to be corrected.
- The authors supplemented the typescript with an inventory table. I would like the information contained in it to be broken down into stages in the assessment process.
- In addition, I am asking for a functional unit.
- The method of assessing the recycling stage is still not clear. The simulation results should be presented for each stage.
I appreciate the authors' hard work, but the manuscript still needs some proofreading.
Author Response
Thank you for reviewing the revised manuscript.
Please see the response to the reviewer's comment in the attached file
